# The Impact of Reduced Training Activity of Elite Kickboxers on Physical Fitness, Body Build, and Performance during Competitions

**DOI:** 10.3390/ijerph18084342

**Published:** 2021-04-20

**Authors:** Tadeusz Ambroży, Łukasz Rydzik, Zbigniew Obmiński, Andrzej T. Klimek, Natalia Serafin, Artur Litwiniuk, Robert Czaja, Wojciech Czarny

**Affiliations:** 1Institute of Sports Sciences, University of Physical Education, 31-571 Krakow, Poland; tadek@ambrozy.pl; 2Department of Endocrinology, Institute of Sport-National Research Institute, 01-982 Warsaw, Poland; zbigniew.obminski@insp.pl; 3Department of Physiology and Biochemistry, Faculty of Physical Education and Sport, University of Physical Education in Kraków, 31-571 Kraków, Poland; andrzej.klimek@awf.krakow.pl; 4Faculty of Physical Education and Sport, Institute of Social Sciences, University of Physical Education in Krakow, 31-571 Kraków, Poland; natalia.ambrozy@gmail.com; 5Faculty of Physical Education and Health in Biała Podlaska, Józef Piłsudski University of Physical Education, 00-809 Warsaw, Poland; a.litwiniuk@wp.pl; 6College of Medical Sciences, Institute of Physical Culture Studies, University of Rzeszow, 35-310 Rzeszów, Poland; rczaja@ur.edu.pl (R.C.); wojciechczarny@wp.pl (W.C.); 7Department of Sports Kinanthropology, Faculty of Sports, Universtiy of Presov, 080-01 Prešov, Slovakia

**Keywords:** physical exercise, COVID-19, pandemic, physical fitness

## Abstract

*Background:* Kickboxing is a combat sport where athletes require constant supervision of a coach. The outbreak of the pandemic caused by SARS-CoV-2 has led to a crisis in many sectors, including sport. Global efforts to limit the spread of the pathogen resulted in temporary lockdowns that limited sporting activity, thus deteriorating athletes’ physical fitness. *Methods:* The participants included 20 kickboxers competing at the international level. Their physical fitness was evaluated based on the test developed by the International Committee on the Standardization of Physical Fitness Test (ICSPFT) and their body build was assessed using Tanita BC601 body composition monitor and a body height meter. Differences between physical fitness before the COVID-19 outbreak and during the pandemic after re-opening sports facilities were evaluated. *Results:* Temporary closing of sports clubs has led to the increase in body mass of the participants by 2.65 kg on average and significantly deteriorated physical fitness. *Conclusions:* Temporary closing of sports clubs and restrictions on physical activity aimed at containing the spread of COVID-19 caused a significant reduction in physical fitness and increased body mass of the participants. This is likely to cause worse performance in the nearest competitions and have a negative impact on the athletes’ health status and immunity to diseases.

## 1. Introduction

The concept of health-related fitness (HRF), which provides the theoretical framework for the development of general fitness, is not always considered in sports practice [1]. Physical fitness is also a clear indicator of human health. Its optimal level, when considering the somatic potential of the athlete, supports the development of special fitness and allows the technical and tactical skills to be improved, which determines the success in any sport. The improvements in physical fitness can be achieved during any training unit and every element of the entire training process [2]. Kickboxing is a relatively new combat sport. A kickboxing fight is based on punching and kicking techniques, with fighters allowed to strike only specified parts of the opponent’s body. The repertoire of the techniques requires adequate physical fitness to use them effectively [3]. Furthermore, maintaining the optimal body mass is an important element of the training process and allows for choosing the weight class in which the competitor performs best [4]. Kickboxing is an individual sport that requires direct coach supervision and consistently following a training process. The coach’s supervision allows the athlete to make faster training progress, and that can result in a greater number of wins [5]. Furthermore, the accessibility of professional sports equipment in sports facilities influences training efficiency. Nowadays, there are better and more precise training tools (punching bags, speed bags, or body opponent bags) [6]. At any stage of the entire process of sports training, maintaining a high level of general fitness is a basic training goal. Properly developed physical fitness underlies the range of kickboxer’s possibilities and it plays a key role in the training process [7]. The definition of fitness given by WHO characterizes it as an important element of health in addition to psychological and social fitness. Such physical fitness is believed to be a source and prerequisite of a full and satisfactory life rather than merely sports successes. According to the definition, physical fitness includes cardiopulmonary function, relative slimness, muscle strength, endurance, and agility. All these elements are directly related to a higher level of quality of life and are meaningful in the prevention of most health problems and increase the body’s immunity [8].

The SARS-CoV-2 virus (Severe Acute Respiratory Syndrome Coronavirus 2), causing the disease called COVID-19, has led to the death of over a million people globally. The COVID-19 outbreak forced governments to introduce severe restrictions and limitations that have negatively influenced many areas of the economy, including the sports sector [9,10,11], and brought about obstacles to outdoor physical activity [12,13]. As long-lasting lower physical activity is harmful to health and physical fitness, some researchers provide recommendations to exercise at home to maintain, at least in part, the previous level of physical capacity [14,15]. However, habitual lifestyles of athletes involve group training sessions, participation in training camps and competitions, which usually enhance their interpersonal relations. The inappropriate level or lack of those social interactions, and difficulties in getting advice from coaches are sources of psychological stress [16]. The temporary closing of the sports clubs forced athletes to follow individual training programs at home [17]. Home-based training requires strong individual motivation and self-control that could interfere with training efficiency.

As mentioned before, success in sport depends on the level of the athlete’s sports skills and his or her physical fitness [18]. The full realization of the training process requires an active coach’s intervention and involvement of the athlete. Regular supervision of the athlete by his or her coach allows better identification of errors made by the athlete and helps improve technical skills [19]. It is possible to define rigorous training routines, but it is controlling the performance of the exercises that causes improved effects [20]. Kickboxing and other martial arts belong to contact sports, in which sparring plays an important role in the development of sport-specific skills. Through regular sparring matches, kickboxers get accustomed to the conditions of sporting competition, gain self-confidence and proper skills that are likely to result in a greater number of wins in the future [21]. It helps athletes familiarize themselves with the values of the sports competition and influences the development of their special and general fitness [22]. Counteracting the COVID-19 pandemic has deprived the fighters of the possibility of live sparring, leaving them with the options of shadowboxing or exercising with a punching bag at home. Shadowboxing improves the speed and flow of movements, but the athletes are unable to practice counterattacks or defensive elements [23]. Studies prove that restrictions on physical activity and the stay-at-home order have led to a significant reduction in the levels of physical activity and an increase in sedentary lifestyles [24]. Due to the prohibition of leaving home and the inability to achieve training goals, the sedentary lifestyle has become much more dominant and has had a strong negative impact on the health and the level of training [25]. Another significant change concerns the psychological health of athletes preparing for competitions at the international level [26]. It should be stressed that for the contestants who want to continue optimal training at home, online advice provided by their coaches might offer a solution to the problem of limited mobility [27].

The objective of this paper was to assess the effects of the COVID-19 lockdown on the physical fitness and body build of kickboxers preparing for the competitions at the international level in an aspect of the influence of the pandemic on the results of the competitions and on the health status of the participants. To verify the sports skill level of the athletes, an analysis of the competitive activity was performed at the beginning and the end of the study. The experiments used in this study were conducted at the time of the outbreak of the COVID-19 pandemic in Poland (March 2020). In view of this situation and the sudden increase in infections, authorities have taken a number of measures aimed to contain the spread of COVID-19. Movement was limited, gatherings were forbidden, and the sporting environment was completely shut down. As a result, professional athletes were deprived of the opportunity to train in professional settings (fully equipped sports halls and direct coaching supervision). Training was limited to that performed in the home environment.

## 2. Materials and Methods

The study included 20 male kickboxers who were preparing for competitions at the international level, with a mean age of 25.2 ± 3.02 years, an average body height of 180.65 ± 3.53 cm, and body mass of 82.5 ± 4.89 kg. They had fought an average of 15 matches a year in national and international competitions, and their training experience depending on their age ranged from 8 to 12 years. The participants were first or master class athletes according to the Polish Kickboxing Association. The frequency of training sessions before the lockdown was 6 times a week, 2 h each. The coaching recommendations for the period of restrictions caused by the pandemic were the same. The athletes were measured for body mass, body height, and physical fitness. Body mass was measured using a Tanita BC-601 body composition monitor (Tanita, Tokyo, Japan), the height was measured using a SECA 2017 body height meter (Seca, Hamburg, Deutschland), and physical fitness was evaluated based on the following ICSPFT tests [26]:Aerobic Capacity Test: evaluation of the human body’s maximum rate of oxygen consumption (VO_2_max, description of the test below).50-meter Sprint (participants run a distance of 50 m as quick as they could).Standing Long Jump (participants jumped from a standing position as far as they could).1000-meter Run (participants run a distance of 1000 m as quick as they could).Grip Strength (participants had to grip the dynamometer as hard as possible with the arm fully extended).Pull Up (participants had to do as many pull-ups as they could in a time given).5 × 10-meter Shuttle Run (participants run a distance of 10 m five times as quickly as they could).Sit-Ups (participants did as many sit-ups as they could in 30 s).Forward Bend (participants had to bend forward and reach the lowest possible point).

The tests were supervised by the authors. Tests 1, 2, 3, and 5 were performed on the first day, whereas tests 4, 6, 7, 8, and 9—on the second day. Two days before the tests, training intensity was reduced to 30–40% of the baseline levels. Participants’ diets were monitored based on the interview and it was found that there were no changes in nutrition during the period of the study. Furthermore, each participant was instructed not to use specialized diets and supplementation during the experiment due to the fact of the high dependence of changes in the body build and composition on diets. The participants did not change their diets during the experiment. Dietary monitoring was based on using notebooks in which the participants recorded the foods they consumed using home measures based on a photo album of foods and products provided. Analysis of nutrition revealed no specialized diet or using performance-enhancing supplements in the study group. The assumed calorific value of consumed foods was derived from the calculations of the total metabolic rate of each participant, which consisted of basal metabolic rate + 10% for digestive processes + calorific value calculated using exercise time and HR measurements during exercise.

The first measurement was performed before the restrictions on access to sporting facilities were imposed on 13 March 2020. The second measurement was carried out on 21 May 2020, after the end of the lockdown and reopening of sports clubs.

Body Mass Index (BMI) was calculated from the measurements of body height and mass.

### 2.1. Measurement of VO_2_max

Maximal oxygen uptake (VO_2_max) was evaluated during the Margaria step test. The participants climbed a 40 cm step. In the first 6-minute stage, the frequency of climbing was 15/min, whereas in the second—25/min. During both stages, heart rate was measured using a heart rate monitor (Polar). Maximal oxygen uptake was computed based on the formula:$${\mathrm{VO}}_{2}\max=\frac{\mathrm{HRmax}\left(\mathrm{VO}2\mathrm{II}-\mathrm{VO}2I\right)+\mathrm{HRII}\times \mathrm{VO}2I\times \mathrm{VO}2\mathrm{II}}{\mathrm{HRII}-\mathrm{HRI}}$$
where:HRmax—maximum heart rate [beats/min.];HRmax computed according to Tanaka 2001 (208−0.7 × age);HRI—heart rate during the first stage [bpm];HRII—heart rate during the second stage [bpm];VO_2_I—estimated oxygen uptake during the first stage [mL/O /kg/min];that requires ca. 2.0 [mL/O /kg/min];VO_2_II—estimated oxygen uptake during the second stage [mL/O /kg/min]that requires ca. 23.4 [mL O_2_ /kg/min].

### 2.2. Measuring the Indices of Technical and Tactical Skill Level

To verify the sports skill level during the tournament, we performed the analysis of the fight and made relevant calculations. The kickboxing bouts were video recorded for further analysis. Next, the indices of a technical and tactical skill level were computed using the following formulas [16,28]:

Efficiency of the attack (Sa)$$\mathrm{Sa}=\frac{n}{N}$$

*N*—numbers of attacks awarded 1 pt.;

In K1 formula, each fair hit is awarded 1 pt.;

*N*—number of bouts.

Effectiveness of the attack (Ea):$$\mathrm{Ea}=\frac{number\;of\;efective\;attacks}{number\;of\;all\;attacks}\times 100$$

An effective attack is a technical action awarded a point.

Number of all attacks is a number of all offensive actions.

Activeness of the attack (Aa):$$\mathrm{Aa}=\frac{number\;of\;all\;recorded\;offensive\;actions\;of\;a\;kickboxer}{number\;of\;bouts\;fought\;by\;a\;kickboxer}$$

### 2.3. Training Procedure

The study group used individual training programs during the lockdown, aimed at maintaining the achieved level of general physical fitness and selected goals of the sport-specific training program. During the home training program, the athletes followed the instructions of the coach, who supervised their progress based on interviews and using remote forms of communication. Despite detailed instructions and the recommendations regarding workload, especially in terms of training intensity, the athletes failed to replicate the previous form of training (before the lockdown), which had been performed under the close supervision of the coach and in training facilities fully equipped with professional training equipment. Another impediment was the lack of contact training performed with a partner.

### 2.4. Bioethical Committee

Prior to participation in the tests, the athletes were informed about the research procedures, which were in accordance with the ethical principles of the World Medical Association Declaration of Helsinki (WMADH, 2000). Obtaining the participant’s written consent was the precondition for their participation in the project. The research was approved by the Bioethics Committee at the Regional Medical Chamber (No. 287/KBL/OIL/2020).

### 2.5. Statistical Analysis

Statistical analysis of the variables was performed using Statistica 13.1 software (StatSoft) (Tibico, Palo Alto, CA, USA). Parametric tests were used depending on meeting the assumptions of normal distribution and homogeneity of variance. The Shapiro–Wilk test was used to test variables for normal distribution. The homogeneity of variance was assessed using Levene’s test. Student’s t-test for dependent samples was used to compare the results recorded in the population at two time points. The correlation of two variables with normal distribution was computed using Pearson’s linear correlation coefficient. The statistical significance level was set at *p* < 0.05.

## 3. Results

Selected parameters of the body build of the participants before the COVID-19 outbreak and after the lockdown were compared. It was found that both body mass and BMI of the athletes increased during the pandemic. The differences were statistically significant (*p* < 0.001) and the effect size was assessed as at least average based on Cohen’s coefficient d (Table 1).

It was shown that the differences between the two measurements were statistically significant in the following tests: Standing Long Jump (*p* < 0.001), 1000-meter Run (*p* = 0.028), Pull Up (*p* < 0.001), 4 × 10-meter Shuttle Run (*p* = 0.015), and Forward Bend (*p* < 0.001). The effect size was the largest for Forward Bend (d = 1.36), but it was also large for Pull Up (d = 0.61). The mean results of all tests after the lockdown were worse than the mean results before the pandemic. Another two results: 50-meter Sprint (*p* = 0.097, d = 0.5) and Grip Strength (*p* = 0.077) were close to being statistically significant. Maximal oxygen uptake (VO_2_max) is considered the main indicator of aerobic capacity. It was somewhat lower after the detraining period in the elite athletes. As shown in Table 2, this variable significantly decreased along with activity, effectiveness, and efficacy levels during the pandemic (Table 2 and Table 3). 

In the majority of cases, the change in BMI had no effect on the change in fitness parameters before and during the pandemic. The relation between the change in BMI and the change in Standing Long Jump was the only one that was statistically significant. The correlation was negative, which means that the higher the increase in BMI during the pandemic, the higher the decrease in the results of Standing Long Jump. Individuals with a lower increase in BMI had a lower decrease in the results of Standing Long Jump (r = −0.45, *p* = 0.045) (Table 4).

## 4. Discussion

The training performed by the athletes studied was based on the classic linear periodization [29]. It was divided into three phases: the preparatory phase (building the sports performance level), the competitive phase (stabilization and using the previously built sports performance level during competitions), and the transition phase (partial and controlled reduction in the sports performance level). In order to optimize the sports performance level, it is necessary to increase the training load in the aspects of training volume and intensity [30]. In the lockdown period, during which the research was carried out, training periodization was disturbed. The coach taking care of the examined athletes had limited possibilities of controlling the training sessions and choosing training loads. It seems necessary to make corrections in the training periodization process after the end of the break caused by the lockdown that disrupted the standard training programs. The introduction of the block periodization [31,32,33] seems to be a reasonable solution, as proposed by the authors. This training modification will allow for compensating for the deficiencies caused by the lockdown. The implementation of a training structure based on block periodization includes periods of shorter duration than in conventional linear periodization (in combat sports, it consists of accumulation for 2 weeks, intensification for 1–4 weeks, and transformation for 2 weeks) [34]. Given the results of our research indicating a significant decrease in the competitive performance of the athletes, such measures seem to be necessary. Remodeling the temporal structure of sports training seems to be a way to return to the optimal competitive performance of the athletes studied [28].

There has to be a wave change. Firstly, training volume is increasing, followed by the increasing intensity with the simultaneous reduction of the volume. A skillful transition from different types of physical exercises to sport-specific training and the transition from aerobic to anaerobic exercise are key elements of the training process. Developing physical fitness, which determines the quality of special training, is also very important. Due to the outbreak of the pandemic and the lack of contact between the coach and the athlete, a full qualitative and quantitative control of the performance of training tasks according to the above principles is impossible. Therefore, the deteriorated development of physical fitness demonstrated in this paper is likely to result in a poorer performance during the upcoming competitions.

Maintaining an appropriate level of physical fitness is a very important element of the athlete’s effectiveness during the kickboxing match [28,30]. Restrictions caused by the COVID-19 outbreak have led to significant changes in the body build and physical fitness of the study participants. The above-mentioned changes were probably caused by closing sports facilities aimed to counteract the COVID-19 pandemic. Low body fat and proper proportions of body build allow the kickboxers to achieve a high level of sports performance [35,36]. Supervised training in a sports club has a big influence on body build. Tremblay et al. reported positive changes in body build after intensive supervised training [37].

A significant increase (2.65 on average) was observed in the body mass of the participants. This is likely to be due to a rapid change in the level of physical activity of the kickboxers following the restrictions on accessibility to professional sports facilities and coach’s supervision imposed during the pandemic. The results of the Chinese study that concerned changes in body mass and lifestyles during the COVID-19 pandemic showed that men with BMI less than 24 gained weight significantly, but those with BMI greater than 24 lost their weight [38]. In this study, the mean BMI before the COVID-19 outbreak was 25.27 and increased during the pandemic to 26.08. Benedini et al. claimed that decreasing physical activity in professional athletes leads to a slowdown of their metabolic functions and results in an increase in their BMI [39]. These results are largely consistent with those obtained in this study. Maintaining proper body mass is a basic prerequisite for participating in a kickboxing competition because it makes it possible for the athlete to compete in a specific weight class [40,41]. Changing a weight class to a heavier one challenges participants to fight with new opponents with different body build proportions. The competitors try to maintain their optimal body weight so they could start in a weight class that fits them best. There was a relation between the increase in BMI and the change in the results in Standing Long Jump. Furthermore, a significant decrease in dynamic strength was observed during the pandemic. Counteracting the spread of COVID-19 significantly decreased the level of physical fitness of the athletes and negatively affected their aerobic capacity levels. It is worth emphasizing that a properly maintained level of aerobic capacity provides an excellent basis for both the performance of high-intensity exercises and the completion of competition tasks [42,43,44]. The greatest decrease was observed in the results of Standing Long Jump, Pull up, and Forward Bend. Statistical significance was also demonstrated in the decrease in the levels of endurance, agility, efficiency, effectiveness, and activity of the players during the bout. A decrease in the technical and tactical skill indices translates into a poorer competitive disposition, which directly translates into sports performance. The Standing Long Jump test measures the explosive strength of lower limbs muscles, which is a very important factor in a kickboxing bout, where kicking techniques are often used. Similarly, the decrease in the strength of the shoulder girdle, reflected in a smaller number of pull-ups could cause a decrease in efficiency. Agility, which allows for performing high kicks easily, is also an important element. The decrease in the results of a 1000-meter Run and Sit-Ups was also statistically significant. Fighting more efficiently and using a greater number of techniques is possible with the appropriate level of endurance. The effect size was the greatest in Forward Bend (d = 1.36) and Pull Up (d = 0.61) tests. The results of all tests during the pandemic were worse than the results before the COVID-19 outbreak. The decrease in all parameters has a significant impact on the results of competitions and the sports performance of kickboxers [35,45]. The monitoring of physical fitness parameters is necessary to ensure success in a competition. Furthermore, the decrease in physical fitness can have negative health consequences. The results of many studies have demonstrated that limited physical activity due to the lockdown can be related to a number of negative metabolic effects that rapidly increase the risk of the occurrence of many serious diseases i.e., diabetes, osteoporosis, cardiovascular diseases, or cancer [46,47,48,49,50,51]. Additionally, limited activity and related decreases in athletes’ fitness can have a negative impact on the psychological domain of both athletes and non-athletes, eventually leading to depression or PTSD symptoms [52,53]. Accumulation of all these factors can result in serious psychological and somatic consequences that could even exclude the participants from future competitions and, according to the HRF concept, can have a negative effect on their health. All these aspects interfere with the athletes’ functioning by reducing the level of competition and training.

Up to date, little is known about the impact of COVID-19 restrictions on athletes’ physical fitness. Our study showed a negative effect of the limited possibility of performing the appropriate training. We suggest that lower training volume was responsible for body weight gain and impaired physical fitness during the pandemic. On the other side, very mild physiological symptoms caused by the infection in some athletes, as reported by mass media, may mask the effects of the lockdown on lower levels or lack physical training. These problems are worth further exploration. The results of the present study are consistent with those published by other researchers, who reported similar impaired physical fitness in athletes during the same period [54,55,56].

## 5. Conclusions

1. The temporary closure of sports facilities and restrictions on physical activity aimed to counteract the COVID-19 pandemic led to a significant decrease in general and special physical fitness of the kickboxers which affected their effectiveness, activity, and efficiency during kickboxing bouts.

2. The restrictions on training organized in sports clubs aimed to counteract the COVID-19 pandemic caused an increase in the body mass of the athletes, consequently affecting their individual weight classes and forcing them to reorganize training programs and diets in order to return to the optimal sports performance.

3. Studies of kickboxers in the period before and after the lockdown periods indicate a number of unfavorable changes occurring in body build, level of physical fitness, and deterioration of the technical and tactical skill level indicators, which may lead to the deterioration of physical and mental health and translate into performance during competitions.

### 5.1. Practical Implications

Taking into account the positive aspects of the restrictions aimed to protect the health and life of the athletes and their environment, and the negative aspects demonstrated in the present study (increase in body mass, decrease in general and special physical fitness), it is necessary to change the training program once the restrictions are lifted towards an accelerated recovery of the optimal level of sports performance and to apply a properly structured training program which should be close to the actual training load. This will allow the athletes to maintain the continuity of their training process with as little deterioration of performance as possible while adhering to the recommended restrictions.

### 5.2. Limitation of the Study

The present study was compiled incidentally due to the sudden outbreak of the COVID-19 pandemic and the need for training in the home environment. The authors had only the results of the presented tests, which were repeated after partial loosening of the lockdown restrictions. It is impossible to carry out a more detailed study because the phase of the pandemic from the period before the outbreak to the loosening of restrictions has already been completed. It is worth noting that the research was conducted on a group of elite athletes, which additionally limits the sample size due to the presented sports skill level.

## Figures and Tables

**Table 1 ijerph-18-04342-t001:** Comparison of the results of body build measurements before and during the pandemic.

Parameters	Before Pandemic		During Pandemic		Mean Difference (95% Cl) During Pandemic–Before Pandemic				Student’s *t*-Test		Effect Size
	m	SD	m	SD	m	SD	−95% Cl	+95% Cl	t	*p*	Cohen *d*
Body mass (kg)	82.50	4.89	85.15	4.87	2.65	1.31	2.04	3.26	9.05	<0.001	0.54
BMI (kg/m^2^)	25.27	1.09	26.08	1.13	0.82	0.41	0.63	1.00	8.99	<0.001	0.75

BMI-Body Mass Index.

**Table 2 ijerph-18-04342-t002:** Comparison of the results of physical fitness measurements before and during the pandemic.

Parameters	Before Pandemic		During Pandemic		Mean Difference (95% CI) During Pandemic–Before Pandemic				Student’s *t*-Test		Effect Size
	m	SD	m	SD	m	SD	−95% CI	+95% CI	t	*p*	Cohen *d*
50-meter Sprint (s)	8.25	0.30	8.41	0.33	0.15	0.39	−0.03	0.34	1.75	0.097	0.50
Standing Long Jump (m)	208.30	15.55	202.70	18.09	−5.60	5.14	−8.01	−3.19	−4.87	<0.001	0.36
1000-meter Run (min)	3.91	0.33	4.06	0.26	0.15	0.28	0.02	0.28	2.38	0.028	0.45
Grip Strength (right hand) (kg)	55.97	2.07	55.72	1.98	−0.25	0.59	−0.52	0.03	−1.87	0.077	0.12
Grip Strength (left hand) (kg)	55.09	2.09	55.05	2.08	−0.05	0.45	−0.26	0.16	−0.47	0.642	0.02
Pull Up (repetitions)	19.70	4.08	17.20	4.23	−2.50	2.16	−3.51	−1.49	−5.16	<0.001	0.61
5 × 10-meter Shuttle Run (s)	10.84	0.69	11.00	0.75	0.16	0.27	0.04	0.29	2.67	0.015	0.23
Forward Bend (cm)	7.25	1.25	5.55	1.10	−1.70	0.80	−2.08	−1.32	−9.49	<0.001	1.36
Sit-Ups (30 s) (repetitions)	27.55	3.91	27.20	4.19	−0.35	1.63	−1.11	0.41	−0.96	0.349	0.09
VO_2_max	47.2	3.7	45.6	3.5	−1.60	2.93	−2.50	−0.70	3.71	0.001	0.496

**Table 3 ijerph-18-04342-t003:** The impact of the pandemic on changes in selected parameters in kickboxers.

Variable	Before Pandemic		During Pandemic		Mean Difference (95% CI) During Pandemic–Before Pandemic				Student’s *t*-Test		Effect Size
	m	SD	m	SD	m	SD	−95% CI	+95% CI	t	*p*	Cohen’s *d*
Activeness	96.9	12.2	90.7	10.8	−6.20	4.74	−8.42	−3.98	5.84	0.000	0.616
Effectiveness	47.5	4.7	44.5	5.1	−3.08	3.68	−4.80	−1.36	3.74	0.001	0.682
Efficiency	50.6	6.6	47.5	3.7	−3.10	5.00	−5.44	−0.76	2.77	0.012	0.586

**Table 4 ijerph-18-04342-t004:** Relation between the change of fitness parameters and the change of BMI.

Variables	Pearson’s
50-meter Run vs. BMI	r = 0.16 *p* = 0.511
Standing Long Jump vs. BMI	r = −0.45 *p* = 0.045
1000-meter Run vs. BMI	r = −0.14 *p* = 0.566
Grip Strength (right hand) vs. BMI	r = 0.22 *p* = 0.345
Grip Strength (left hand) vs. BMI	r = −0.23 *p* = 0.336
Pull Up vs. BMI	r = −0.08 *p* = 0.738
4 × 10-meter Shuttle Run vs. BMI	r = 0.22 *p* = 0.353
Forward Bend vs. BMI	r = 0.09 *p* = 0.719
Sit-Ups vs. BMI	r = 0.12 *p* = 0.603
VO2max vs. BMI	r = 0.05 *p* = 0.848
Activity vs. BMI	r = −0.05 *p* = 0.079
Effectiveness vs. BMI	r = 0.21 *p* = 0.379
Efficacy vs. BMI	r = −0.26 *p* = 0.278

## Data Availability

The data presented in this study are available on request from the corresponding author.

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
