# Peer review of "The Impact of Reduced Training Activity of Elite Kickboxers on Physical Fitness, Body Build, and Performance during Competitions"

_ijerph, 2021, doi:10.3390/ijerph18084342_

Round 1

Reviewer 1 Report

The present study analyzed the effects of lockdown on kickboxer body composition and performance. It is one more study about lockdown consequences on physical fitness, this time focusing on kikboxing athletes. 
It is interensting to ascertain how this specific population managed exercise training during lockdown, however, I miss more information about this in the manuscript. Authors must provide specific information about athletes training during study, further, because they assume that body mass incresed becuased of it. Also, diet data must be provided. Authors state that food ingestion did not change during the study, but, because food ingestion may be relevant in body mass changes, they must show data about it. 

Authors declare that "first measurement was done before introducing the restrictions in sports activities". Please, state the exact date, because restrictions were set at different dates depending on the country. 

I miss information about sex, age, experience and competitive level of the athletes.

Since the Tanita BC-601 scale was used, why don't the authors present data on body composition, water content, etc. Although the information is not comparable to what a DEXA analysis can offer, it is relevant because it helps to determine if the weight gain is muscle or fat. In this line, impedance is very influenced by hydration status. ¿Did the authors control this variable? 

I would not expect height to change, unless athletes were still growing up (one reason to inform about athletes age). I rather delete these data from table 1 and inform about it with descriptive data (age, sex,...).

I am not an expert in Margaria Test, but VO2max data reported seems low for competitive athletes to me. Were they unfit or may the test underestimate VO2max data?

In the first paragraph of the discussion, authors declare that training process is set in three main periods, coinciding with the classical linear periodization. There are more training periodization models susceptible, and may me more effective, to be applied in fight sports. Further, reference 28 refers to COPD patients. I am sure authors can find more adequate references about training process and kickboxing.

Regarding conclusions and practical implications:
I do not agree with the authors, I see that performance decreased during lockdown, but I do not believe the decrease reported is relevant for competition. Further, as in any sport in any country, after lockdown, all sport competitions were rearranged and and set new dates, in order to allow athletes to optimize performance.  
Besides, authors declare that "temporary closing of sports clubs" (this does not mean that athletes stop training) could have a negative impact in health status and immunity. This assumption seems speculative to me, since authors did not assessed health (except VO2 and body weight) and immunity. Same for the alleged negative effects on a psychological level. I do not mean that perforamnce decreased observed is not an important finding, but the authors throw too many speculations and assumptions about its consequences.
Finally, the practical implication indicated questions the public health benefits taken during confinement over the harm caused to kickboxing athletes (by the way, social distancing in not a positive aspect of the restrictions, it is a restriction that causes positive aspects, such as protecting people's health and lives). I believe the policy of introducing restrictions in professional sports is clearly enough. Moreover, today professional sport is a fairly protected and well managed activity by administrations in order to ensure its operation limiting the risk of contagion and protecting the health of athletes.

Manuscript need a deep revision of english grammar. Reading is not very uncomfortable, but there are serious writing mistakes.

Author Response

Dear Reviewer,

Thank you very much for your time and valuable comments, which all have been considered and incorporated. The detailed list of responses is given below. We hope that the modifications and explanation will be acceptable for you.

Yours sincerely,

Rydzik, corresponding author

It is interensting to ascertain how this specific population managed exercise training during lockdown, however, I miss more information about this in the manuscript. Authors must provide specific information about athletes training during study, further, because they assume that body mass incresed becuased of it. Also, diet data must be provided. Authors state that food ingestion did not change during the study, but, because food ingestion may be relevant in body mass changes, they must show data about it. 

A: The description of the training procedure during the lockdown has been added and the information on diets and how they were assessed has been extended.

Authors declare that "first measurement was done before introducing the restrictions in sports activities". Please, state the exact date, because restrictions were set at different dates depending on the country. 

A: Dates have been added.

I miss information about sex, age, experience and competitive level of the athletes.

A: The relevant information has been added.

Since the Tanita BC-601 scale was used, why don't the authors present data on body composition, water content, etc. Although the information is not comparable to what a DEXA analysis can offer, it is relevant because it helps to determine if the weight gain is muscle or fat. In this line, impedance is very influenced by hydration status. ¿Did the authors control this variable? 

A: We do not have detailed data from the first experiment on water and body fat levels. Therefore, these variables were not measured in the second experiment.

I would not expect height to change, unless athletes were still growing up (one reason to inform about athletes age). I rather delete these data from table 1 and inform about it with descriptive data (age, sex,...).

A: Body height has been removed at the request of the Reviewer, but it was left in the calculations of BMI.

I am not an expert in Margaria Test, but VO2max data reported seems low for competitive athletes to me. Were they unfit or may the test underestimate VO2max data?

A: In the Margaria test used, VO2max varied between an average of 47.2 in the first experiment and 45.6 in the second. Compared to the representatives of other combat sports such as judo, boxing, or MMA, this result is similar, which was also reported in the paper “Rydzik, Ł.; Ambroży, T. Physical fitness and the level of technical and tactical training of kickboxers. Int. J. Environ. Res. Public Health 2021, 18, 1–9, doi:10.3390/ijerph18063088.” Furthermore, kickboxing is characterized by a short duration of bouts and therefore such VO2max levels are sufficient.

In the first paragraph of the discussion, authors declare that training process is set in three main periods, coinciding with the classical linear periodization. There are more training periodization models susceptible, and may me more effective, to be applied in fight sports. Further, reference 28 refers to COPD patients. I am sure authors can find more adequate references about training process and kickboxing.

A:  The first paragraph of the discussion has been complemented and expanded and the literature items have been added and revised.

Regarding conclusions and practical implications:
I do not agree with the authors, I see that performance decreased during lockdown, but I do not believe the decrease reported is relevant for competition. Further, as in any sport in any country, after lockdown, all sport competitions were rearranged and and set new dates, in order to allow athletes to optimize performance.  

A: We would like to thank you for this comment, the conclusions have been re-designed and extended. However, we would like to point out that in our research, apart from a decrease in general fitness, we also noticed a deterioration in the indices of special (technical and tactical) fitness, which are directly related to the effectiveness, activity, and efficiency of the fights during competition. Therefore, we found an adverse effect of the break in standard training on the competitive performance of the athletes. We have taken into account the change of competition dates and that is why we have supplemented the practical implications by suggesting a modification of the time structure towards block periodization. 

Besides, authors declare that "temporary closing of sports clubs" (this does not mean that athletes stop training) could have a negative impact in health status and immunity. This assumption seems speculative to me, since authors did not assessed health (except VO2 and body weight) and immunity. Same for the alleged negative effects on a psychological level. I do not mean that perforamnce decreased observed is not an important finding, but the authors throw too many speculations and assumptions about its consequences.

A: Our study indicates a reduction in the level of training loads due to temporarily closing sports clubs. We did not examine the associations between health status and deterioration of physical fitness. Therefore, the conclusions were modified. It seems, however, that lowering the competitive performance and fitness level may have a negative impact on the mental health of the athletes. A decline in physical abilities of the athlete caused by external factors can be compared to an incidental injury before the participation in the competition and such a situation has a negative impact on the mental health of athletes.

Finally, the practical implication indicated questions the public health benefits taken during confinement over the harm caused to kickboxing athletes (by the way, social distancing in not a positive aspect of the restrictions, it is a restriction that causes positive aspects, such as protecting people's health and lives). I believe the policy of introducing restrictions in professional sports is clearly enough. Moreover, today professional sport is a fairly protected and well managed activity by administrations in order to ensure its operation limiting the risk of contagion and protecting the health of athletes.

A: We agree with the comment and have made the modifications. However, our research concerns the initial phase of the pandemic in Poland and the introduction of a full lockdown, which was not thoroughly developed and well-thought by politicians at the time. Current measures in this field are definitely more precise and more oriented towards protecting the health of athletes and their environment and, as far as possible, facilitating the implementation of the training process. We wrote the third conclusion using the conditional mood and its assumptions are based on the literature review:

Fox-Harding, C., Harris, S. A., Rogers, S. L., Vial, S., Beranek, P., Turner, M., & Cruickshank, T. (2021). A survey to evaluate the association of COVID-19 restrictions on perceived mood and coping in Australian community level athletes.

Turgut, M., Soylu, Y., & Metin, S. N. (2020). Physical activity, night eating, and mood state profiles of athletes during the COVID-19 pandemic. Progress in Nutrition, 22.

Berlin, A. A., Kop, W. J., & Deuster, P. A. (2006). Depressive mood symptoms and fatigue after exercise withdrawal: the potential role of decreased fitness. Psychosomatic medicine68(2), 224-230.

Manuscript need a deep revision of english grammar. Reading is not very uncomfortable, but there are serious writing mistakes

A: The study has been edited and corrected for grammar and language.

Reviewer 2 Report

The manuscript "The impact of reducing training activity of elite kickboxers on the level of fitness, body built and the efficiency in competitions" is relevant to the specific area, mainly due to the moment we live with COVID-19. Thus, research involving athletes in this period is very important for adaptation and understanding related to training / competitions. I write below some suggestions for improvements in the text.

I suggest that the authors describe the moment of the pandemic that was in the country of the collected athletes, presenting the restrictions of the moment. Making this clear is important because in each country the restrictions occurred in a specific way. Also if the athletes in the study continued to train at home.

In Materials and Methods, why were tests 1,2,3,5 performed on the first day and 4,6,7,8,9 on the second? How was that choice of tests for each day? Why did they do it this way? Explain.

I suggest that the authors add a paragraph with the sample information, such as age of the athletes, experience with the modality, level, level of the competition they were preparing to participated, number of training sessions carried out before and during the pandemic.

Finally, I would like the authors to add suggestions to try to reduce these losses during the restrictions period, related to kickboxing athletes.

Author Response

Dear Reviewer,

Thank you very much for your time and valuable comments, which all have been considered and incorporated. The detailed list of responses is given below. We hope that the modifications and explanation will be acceptable for you.

Yours sincerely,

Rydzik, corresponding author

I suggest that the authors describe the moment of the pandemic that was in the country of the collected athletes, presenting the restrictions of the moment. Making this clear is important because in each country the restrictions occurred in a specific way. Also if the athletes in the study continued to train at home.

A: We have added information about restrictions in Poland and training. The description of the training procedure during the lockdown has been complemented in the methodology.

In Materials and Methods, why were tests 1,2,3,5 performed on the first day and 4,6,7,8,9 on the second? How was that choice of tests for each day? Why did they do it this way? Explain

A: The division into two test days was due to the care for reliable execution of all tests by avoiding the effect of fatigue. Therefore, the fitness test, speed test, and selected strength tests were carried out on the first day, with the time intervals allowing for a full recovery. On the second day, endurance was measured and other tests of speed, strength, and flexibility were performed following the same principles. The final experiment was carried out based on the same principles, which allowed for maintaining reliability, reproducibility, and accuracy of measurement, and offered the opportunity to compare the results.

I suggest that the authors add a paragraph with the sample information, such as age of the athletes, experience with the modality, level, level of the competition they were preparing to participated, number of training sessions carried out before and during the pandemic.

A: This part has been complemented

Finally, I would like the authors to add suggestions to try to reduce these losses during the restrictions period, related to kickboxing athletes.

A: Relevant information has been added 

Round 2

Reviewer 1 Report

I want to thanks the authors for such a good work answering my concerns. However, I still have some minor questions I would like to be answered:

_ Lines 122-123: please, indicate age, body height and body mass as mean ± SD

_ Authors state in the discussion and in the practical implications sections that block periodization is needed to reverse the negative effects on performance during a restrictive situation such as covid19 lockdown. However, procedures were made to assesse the effect of that lockdown on athletes, not to evaluate an specific training periodization. Unless authors declare it better in the methods section, and introduce the knowledge of block peridization, it seems circunstantial to me how coaches adapted the training programs. I mean, this was not an excersice intervention study, right? By the way, were all training plannings modified under the same parameter for all atheltes?

_ Lines 368 to 371: If possible, it is better to refer to studies that have showed negative metabolic effects due to covid19 lockdown specifically. 

Author Response

Dear Reviewer,

Thank you very much for your time and valuable comments, which all have been considered and incorporated. The detailed list of responses is given below. We hope that the modifications and explanation will be acceptable for you.

Yours sincerely,

Rydzik, corresponding author

_ Lines 122-123: please, indicate age, body height and body mass as mean ± SD

The relevant information has been added following the Reviewer's suggestions.

Authors state in the discussion and in the practical implications sections that block periodization is needed to reverse the negative effects on performance during a restrictive situation such as covid19 lockdown. However, procedures were made to assesse the effect of that lockdown on athletes, not to evaluate an specific training periodization. Unless authors declare it better in the methods section, and introduce the knowledge of block peridization, it seems circunstantial to me how coaches adapted the training programs. I mean, this was not an excersice intervention study, right? By the way, were all training plannings modified under the same parameter for all atheltes?

A: Thank you very much for these valuable suggestions. Obviously, the purpose of the study was to evaluate the effects of lockdown on physical fitness and body composition. The sentence concerning training periodization was removed from the practical implications section. However, we would like to point out that we made a correction in the part of the discussion, concerning not the need but rather the possibility (the sentences are now written in the conditional mood to sound as a proposal) of an accelerated periodization as a way of compensating for the shortages caused by lockdown, which are shown by the results of our research. We would like to mention that one of the authors of this paper is a coach of the examined players and implements the proposed changes in the training structure. The results concerning physical fitness, body build, and the level of technical and tactical training will be presented in the next publication after the completion of the training process.  We would like our study to have practical applications that would be interesting for coaches affected by the pandemic and help them bring players to the right level of training. Based on the results of our own research, which illustrate the unfavorable trends caused by lockdown and on the literature, we attempted to present suggestions for possible solutions to this situation.

_ Lines 368 to 371: If possible, it is better to refer to studies that have showed negative metabolic effects due to covid19 lockdown specifically.

A: We have made additional reference to the studies on metabolic effects.

Martinez-Ferran, M.; de la Guía-Galipienso, F.; Sanchis-Gomar, F.; Pareja-Galeano, H. Metabolic Impacts of Confinement during the COVID-19 Pandemic Due to Modified Diet and Physical Activity Habits. Nutrients 2020, 12, 1549, doi:10.3390/nu12061549.

Jakobsson, J.; Malm, C.; Furberg, M.; Ekelund, U.; Svensson, M. Physical Activity During the Coronavirus (COVID-19) Pandemic: Prevention of a Decline in Metabolic and Immunological Functions. Front. Sport. Act. Living 2020, 2, doi:10.3389/fspor.2020.00057.